

# Predicting the potential distribution of the endemic seabird *Pelecanus thagus* in the Humboldt Current Large Marine Ecosystem under different climate change scenarios

Jaime A. Cursach[1], Aldo Arriagada[2], Jaime R. Rau[3], Jaime Ojeda[4,5,6], Gustavo Bizama[7] and Anderson Becerra[8]

[1] Programa de Doctorado en Ciencias Mención Manejo y Conservación de Recursos Naturales, Universidad de Los Lagos, Puerto Montt, Chile
[2] Laboratorio de Limnología, Departamento de Acuicultura y Recursos Agroalimentarios, Universidad de Los Lagos, Osorno, Chile
[3] Laboratorio de Ecología, Departamento de Ciencias Biológicas & Biodiversidad, Universidad de Los Lagos, Osorno, Chile
[4] Laboratorio de Ecosistemas Marinos Antárticos y Subantárticos (LEMAS), Universidad de Magallanes, Punta Arenas, Chile
[5] Instituto de Ecología y Biodiversidad (IEB), Santiago, Chile
[6] School of Environmental Studies, University of Victoria, Victoria, British Colombia, Canada
[7] Facultad de Ciencias, Universidad de Chile, Santiago, Chile
[8] Programa de Master en Ciencias y Tecnología Espacial, Escuela de Ingeniería, Universidad del País Vasco, Bilbao, Spain

Corresponding author
Jaime A. Cursach, jcurval@gmail.com

## ABSTRACT

**Background**. The effects of global climate change on species inhabiting marine ecosystems are of growing concern, especially for endemic species that are sensitive due to restricted distribution. One method employed for determining the effects of climate change on the distribution of these organisms is species distribution modeling.
**Methods**. We generated a model to evaluate the potential geographic distribution and breeding distribution of the Peruvian pelican (*Pelecanus thagus*). Based on maximum entropy modeling (MaxEnt), we identified the environmental factors that currently affect its geographic distribution and breeding. Then we predicted its future distribution range under two climate change scenarios: moderate (rcp 2.6) and severe (rcp 8.5).
**Results**. The mean daytime temperature range and marine primary productivity explain the current potential distribution and breeding of the pelican. Under the future climate change scenarios, the spatial distribution of the pelican is predicted to slightly change. While the breeding distribution of the pelican can benefit in the moderate scenario, it is predicted to decrease (near −20 %) in the severe scenario.
**Discussion**. The current potential geographic distribution of the pelican is influenced to a large extent by thermal conditions and primary productivity. Under the moderate scenario, a slight increase in pelican breeding distribution is predicted. This increase in habitable area is explained by the climatic conditions in southern Chile, and those climatic conditions will likely be similar to the current conditions of the central coast of Chile. We predict that the coasts of southern Chile will constitute an important refuge for the conservation of the Peruvian pelican under future climate change scenarios.

## INTRODUCTION

Climate change is of increasing concern for seabirds because it negatively affects their conservation status and has become the third most important threat after exotic invasive species and incidental capture (*Croxall et al., 2012*). In turn, a great proportion of seabirds (e.g., of the Humboldt Current System) feed in a relatively narrow range of trophic levels, mainly on larger zooplankton, small pelagic fish, or squid (*Quillfeldt & Masello, 2013*). Most of the prey species consumed by seabirds are strongly affected by climate-induced changes on the productivity of phytoplankton, generating changes in both the abundance and fecundity of herbivorous zooplankton (small copepods and euphausiids). Consequently, carnivorous zooplankton and pelagic fish or squid are also affected (*Crawford et al., 2008a*; *Crawford et al., 2008b*; *Wynn et al., 2007*; *Luczak et al., 2011*). The dynamics of small pelagic fish have been studied intensively in the marine upwelling ecosystems such the Humboldt and Benguela currents, where the collapse of small populations of pelagic fish is often followed by severe decreases in the populations of seabirds (*Crawford & Jahncke, 1999*; *Crawford et al., 2008a*). Seabirds face multiple imminent threats (overfishing and incidental death, pollution, introduced species, habitat destruction, and human disturbance) that may seem more urgent than gradual climate change and its associated climate phenomena (*Croxall et al., 2012*; *Quillfeldt & Masello, 2013*). However, some of these threats are locally restricted, whereas the climate phenomena have the potential to alter an entire region and increase the cumulative pressures that affect many seabirds, especially endemic species (*Quillfeldt & Masello, 2013*; *Jenouvrier et al., 2014*).

The Peruvian pelican *Pelecanus thagus* (hereafter pelican) is a seabird endemic to the Humboldt Current Large Marine Ecosystem (HCLME) of South America. The pelican's home range lies on the Pacific coast from southern Ecuador, through Peru down to southern Chile (*BirdLife International, 2018*). However, it breeding distribution is not continuous along the coast, but is very localized in certain coastal islands from Santa Clara Island (3°S) in southern Ecuador, to Mocha Island (38°S) in central Chile (*Housse, 1945*; *Vinueza, Sornoza & Yáñez Muñoz, 2015*). At the global level, the pelican is classified as near threatened (*BirdLife International, 2018*). In Peru, this species is considered endangered (*MINAGRI, 2014*). In Chile and Ecuador there is no classification concerning its conservation status, even though the Chilean coastline comprises more than 50% of pelican's habitat range (*Cursach et al., 2018*). Between 2010 and 2015 the abundance of pelicans in Chile decreased significantly on the central coast, area encompasses the main breeding population (*Cursach et al., 2018*).

Predicting the response of biodiversity to climate change has developed into an active field of research (*Bellard et al., 2012*; *Molinos et al., 2015*; *Pecl et al., 2017*). Therefore, projections of species distribution models play an important role in alerting scientists and decision makers to assess the potential future risks of climate change (*Pereira et al., 2010*; *Parmesan et al., 2011*). Climate change may alter the suitability of habitat and contraction

of the distribution range of several groups of marine and terrestrial organisms, including Southern Ocean seabirds (*Marzloff et al., 2016*; *Krüger et al., 2018*). The current study aims to generate models of the potential geographic distribution and breeding of the pelican, to identify the environmental factors that affect its current distribution, and to predict its future distribution range under two climate change scenarios (moderate and severe). Our hypothesis was that the spatial distribution and breeding distribution of the pelican will decrease and that the main cause of this will be climate change.

## MATERIALS & METHODS

### Species records

Pelican nesting and occurrence data were compiled from four main sources: the Neotropical Waterbird Census (https://lac.wetlands.org/), eBird (https://ebird.org/), the Global Biodiversity Information Facility (https://www.gbif.org/), and the literature. The geo-coordinates for each data point were referenced from the information in the literature or through the use of coordinates in Google Earth. We excluded duplicate or unclear locations and verified the accuracy of the data. We found a total of 4,818 georeferenced data points referring to pelican sightings (in resting place, nesting sites, coves, beaches, etc.), encompassing its entire geographic distribution from 2000 to 2015. Of these records, a subsampling was performed at a distance of 15 km (cell size), obtaining a total of 264 records, with which the modeling was performed. This subsampling were conducted in R, version 3.0.2 (*R Development Core Team, 2013*). The breeding distribution of the pelican was modeling with information for 34 nesting sites (*Vinueza, Sornoza & Yáñez Muñoz, 2015*; *Zavalaga, 2015*; *Cursach et al., 2018*).

### Environmental variables

The environmental variables used to characterize the current distribution (and breeding) of the pelican were selected based on climate and oceanography. The climate variables used in this study were downloaded from the EcoClimate database (http://www.ecoclimate.org) (*Lima-Ribeiro et al., 2015*). These variables were represented by maximum, minimum, and mean values of monthly, quarterly, and annual temperatures, and the precipitation values recorded between 1950 and 2000. These parameters provided a combination of means, extremes, and seasonal differences in variables known to influence the distribution of species (*Root et al., 2003*). With the species distribution modeling toolbox extension implemented in ArcGIS, all bioclimate variables that showed a correlation higher than 0.7 were eliminated (*Brown, 2014*). Finally, six climate variables were selected: annual mean temperature, mean daytime temperature range, isothermality, seasonality in temperature, annual precipitation, seasonality in precipitation. The oceanographic variables used were sea surface temperature (SST) and marine net primary productivity (mg C m$^{-2}$ day$^{-1}$), as they are considered the main descriptors of the spatial distribution of seabirds (*Quillfeldt et al., 2015*; *Ingenloff, 2017*). These variables were obtained from the National Oceanic and Atmospheric Administration (NOAA, http://www.ngdc.noaa.gov/). For the analyses, we used mean values per climate season for a period of nine years (2004 to 2013), totaling eight oceanographic variables. All environmental variables used in this study were interpolated

by the kriging method, with a uniform resolution of 0.5° × 0.5° using the QGIS 3.2.0 software (*Lima-Ribeiro et al., 2015*; *Varela, Lima-Ribeiro & Terribile, 2015*).

To evaluate the effects of the different climate change scenarios on the spatial distribution of pelicans, we did not include the oceanographic variables. The future climate scenarios corresponded to those proposed by the Intergovernmental Panel on Climate Change (*IPCC, 2014*). These scenarios were obtained from the ecoClimate website (http://ecoclimate.org/), which contains climate models available for different temporal intervals. To do this, we used the model developed by the Community Climate System Model version 4 of the National Center for Atmospheric Research (*Gent et al., 2011*). This is due to the good results for the South-East Pacific (*Larson, Pegion & Kirtman, 2018*; *Zheng et al., 2018*).

The projections for the six preselected variables and the projected minimum and maximum trajectories of the concentrations of greenhouse gases were obtained. That is 2.6 and 8.5 rcp (representative concentration pathways), respectively. These values indicate increases in the heat absorbed by the planet Earth due to the concentration of greenhouse gases up to 2100, in each trajectory and expressed in watts per square meter. Thus, 2.6 rcp is the moderate projection for the scenario with the least climate change, whereas 8.5 rcp is a more pessimistic projection and represents a severe scenario with the greatest climate change (*Taylor, Stouffer & Meehl, 2012*).

## Modeling of the potential geographic distribution

The MaxEnt software (MaxEnt version 3.3.3k, http://www.cs.princeton.edu/~schapire/maxent/) has been frequently used for species distribution models under current and future climate scenarios (*Phillips & Dudík, 2008*). We used MaxEnt to model the geographic distribution of the pelican, including under two previously described climate change scenarios (*Elith et al., 2006*; *Taylor, Stouffer & Meehl, 2012*). The model was elaborated by MaxEnt auto-features (5,000 iterations). Logistic output was used for all analyses. The quality of the model was evaluated using the area under the curve (AUC) and the continuous Boyce index (*Hirzel et al., 2006*). AUC values can vary from 0 to 1, where a value greater than 0.9 is considered an indicator of "good" discrimination skills (*Peterson et al., 2011*). Values of the Boyce index vary between −1 and 1, where positive values indicate a model with predictions that are consistent with the distribution of observed presences in the evaluation dataset (*Boyce, 2002*). Both analyses were conducted in R using the "biomod2" package (*R Development Core Team, 2013*).

For each distribution model, a 30-fold cross-validation was used, with a data proportion of 25% for training and 75% for evaluation. The most important environmental variables were identified by estimating the relative contribution (%) to the model (*Phillips, Anderson & Schapire, 2006*). Jackknife test was used to evaluate the importance of the environmental variables for predictive modeling (*Almalki et al., 2015*).

## RESULTS

### Model yield for potential distribution

The model of presence with the best fit showed a gain of 3.04 and a Boyce Index of 0.99. Also, an $AUC_{training}$ of 0.98 and an $AUC_{evaluation}$ of 0.98 and a standard deviation of 0.004.

**Table 1  Probability of occurrence ranges of the Peruvian Pelican (*Pelecanus thagus*) expressed in surface area.**

| Potential geographic distribution | | Potential reproductive distribution | |
|---|---|---|---|
| Probability of occurrence | Projected surface (km²) | Probability of occurrence | Projected surface (km²) |
| 0.16–0.25 | 174,841 | 0.1–0.2 | 103,148 |
| 0.25–0.33 | 82,153 | 0.2–0.3 | 49,407 |
| 0.33–0.42 | 40,498 | 0.3–0.4 | 63,245 |
| 0.42–0.50 | 59,119 | 0.4–0.5 | 31,296 |
| 0.50–0.59 | 43,793 | 0.5–0.6 | 28,232 |
| 0.59–0.67 | 36,910 | 0.6–0.7 | 110,200 |
| 0.67–0.76 | 18,950 | 0.7–0.8 | 88,326 |
| 0.76–0.84 | 10,572 | 0.8–0.9 | 0 |
| **Total** | **466,836** | **Total** | **473,854** |

While the modeling of breeding distribution showed a gain of 2.24 and a Boyce Index of 0.98, with an $AUC_{training}$ of 0.98 and an $AUC_{evaluation}$ of 0.98 and a standard deviation of 0.003. The AUC values were relatively similar, so the models used are appropriate for predicting the presence and breeding distribution of the species. $AUC_{evaluation}$ 0.98 indicates that the pelican has a wide geographic distribution and breeding in relation to the area corresponding to the environmental data. The model predicts that the potential geographic distribution of the pelican reaches an approximate surface area of 466,836 km², latitudinally distributed from southern Ecuador (2°13′09″S) to southern Chile (46°59′07″S). Over this extensive marine–coastal surface, the probability of occurrence for this species varied between a 0.16 (minimum) and 0.84 (maximum) (Table 1). Areas with the highest probabilities of occurrence for the pelican are represented with intense red colors in Figs. 1A and 2A. These areas are mainly distributed from northern Peru to central Chile.

## Importance of environmental variables

Among the six climatic variables and eight oceanographic variables, the mean daytime temperature range (Bio2) and the summer marine primary productivity, contributed the most to the current and potential distribution of the pelican (Table 2). These two factors explained 78.47% of the modeled distribution. The mean daytime temperature responded to the probability of the presence of the pelican, with a high probability of finding the species in areas where the mean daytime temperature ranges between 6 and 8 °C. In turn, the summer marine primary productivity also influenced the probability of the presence of the pelican, with a greater probability of finding the species during the summer season in areas with high primary productivity. The other factors such as, spring marine primary productivity, isothermality, and seasonality in temperature, contributed 9.24%, 3.23%, and 1.74%, respectively, to the modeled distribution. Therefore, thermal and primary productivity conditions are more important than other variables for mapping pelican distribution (Table 2).
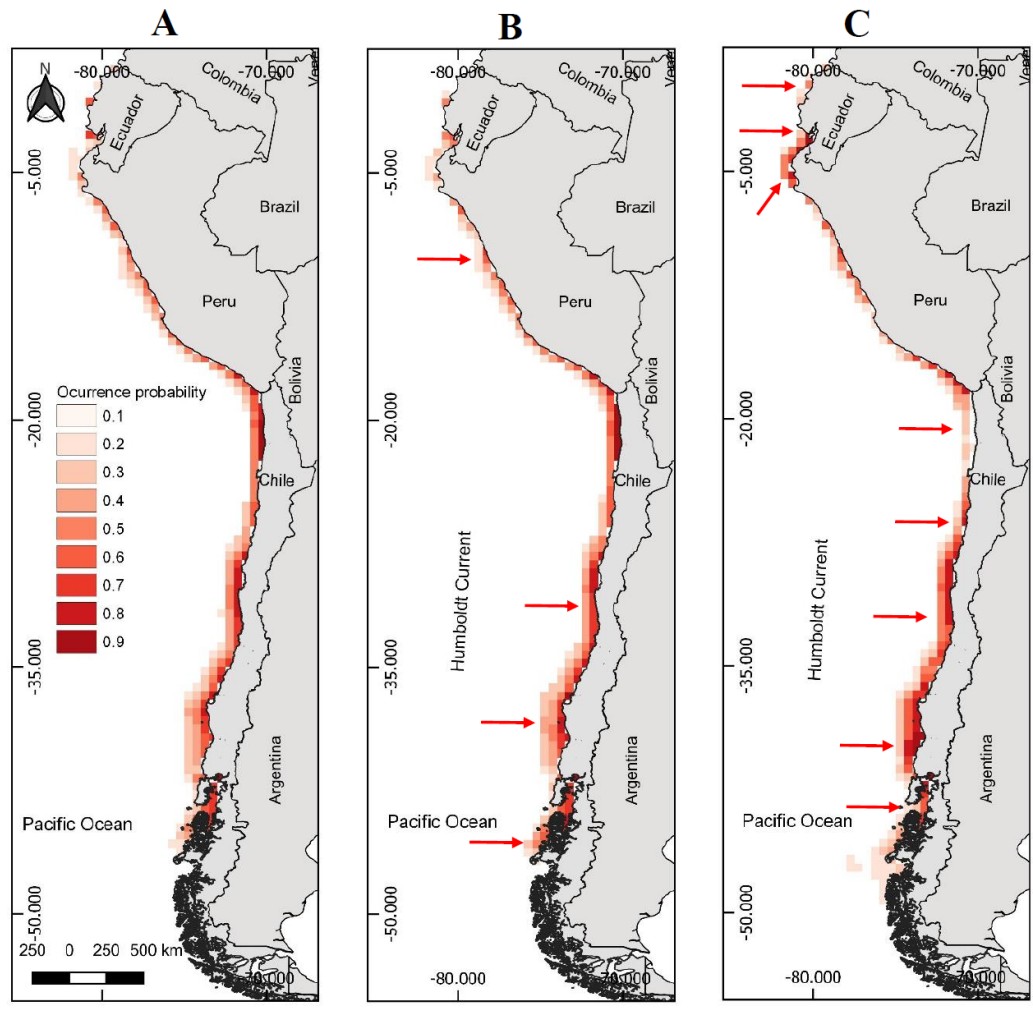

**Figure 1 Models of potential geographic distribution of the Peruvian pelican (*P. thagus*) based on climatic variables and projected for 2010 according to two climate change scenarios.** (A) Projection of current geographic distribution; (B) Projection at 2.6 rcp; (C) Projection at 8.5 rcp. The arrows show relative change to the current distribution.

The modeling of breeding distribution showed that the mean daytime temperature range contributed with 91.5% to the model, while the summer marine primary productivity contributed with 8.5%.

## Potential geographic distribution of the pelican as a function of climate change

Based on the six climatic variables selected in the study, the model predicts that the projected pelican distribution currently attains an area of 596,753 km$^2$ (Table 3). This area is larger than that initially projected (466,836 km$^2$), where the oceanographic variables were integrated. Regarding the projections of climate change for 2100, under the moderate scenario of 2.6 rcp a slight decrease (−0.68%) in pelican spatial distribution is predicted

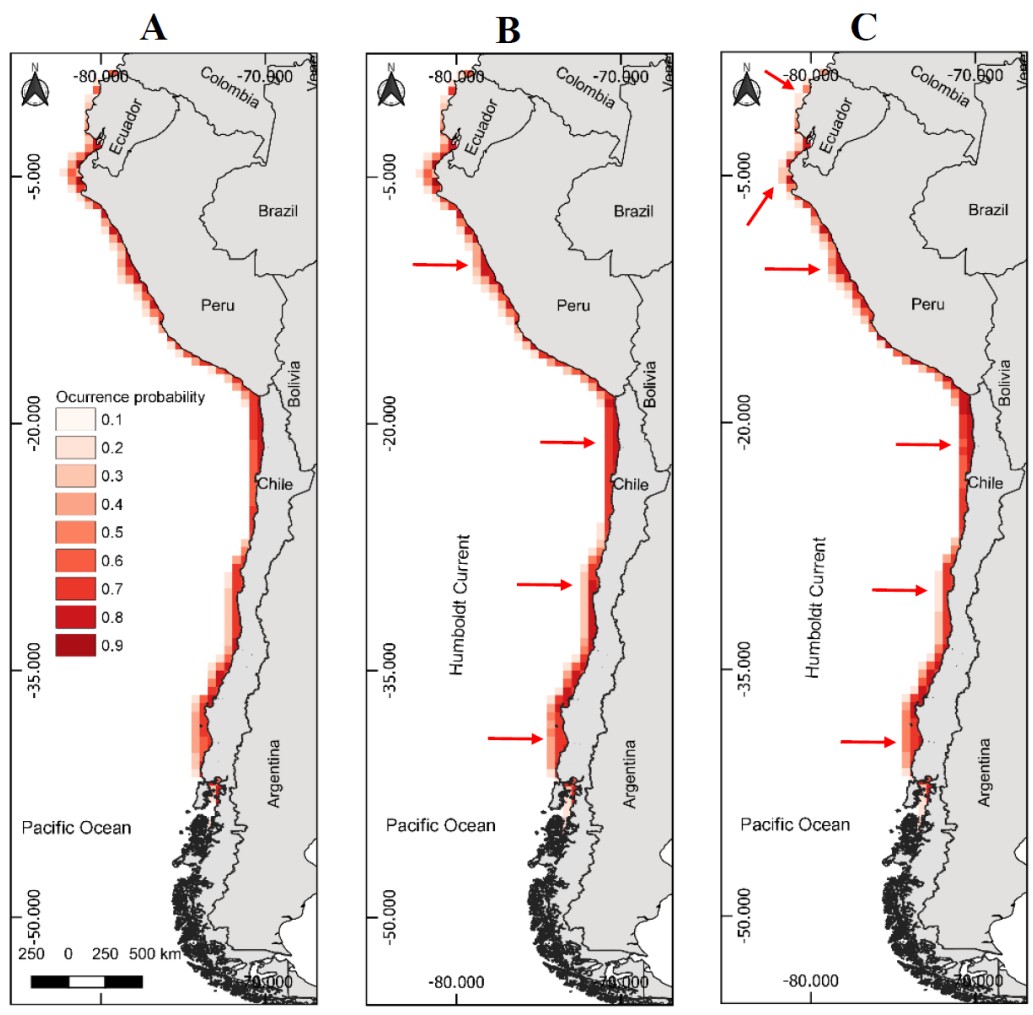

**Figure 2** Models of potential reproductive distribution of the Peruvian pelican (*P. thagus*) based on climatic variables and projected for 2010 according to two climate change scenarios. (A) Projection of current reproductive distribution; (B) projection at 2.6 rcp; (C) projection at 8.5 rcp. The arrows show relative change to the current distribution.

(Table 3). Under the severe scenario of 8.5 rcp, a slight increase (4.51%) in pelican spatial distribution is predicted (Table 3).

The projected habitable surface area under climate change of 2.6 rcp does not presents a major change with respect to the current geographic distribution of the pelican (Table 3). Under the severe scenario, the model predicts that the pelican habitable surface will vary depending on geographic area (Fig. 1). For example, in northern Chile its habitable surface would decrease, whereas in central and southern Chile it would increase over time (Fig. 2). The projected habitable surface area and the probabilities of occurrence for the pelican are spatially schematized in Fig. 2.

For the case of the modeling of breeding distribution, an area of 435,640 km² is projected (Table 4). Regarding the projections of climate change for 2100, under the

**Table 2** Contribution of environmental variables to the current potential distribution model of the Peruvian pelican (*Pelecanus thagus*).

| Variable | Contribution to the model (%) | Importance in permutation (%) |
|---|---|---|
| Mean daytime temperature range | 46.03 | 12.28 |
| Summer marine primary productivity | 32.44 | 1.26 |
| Spring marine primary productivity | 9.24 | 41.20 |
| Isothermality | 3.23 | 0.02 |
| Seasonality in temperature | 1.74 | 0.61 |
| Sea surface temperature in winter | 1.47 | 0.90 |
| Sea surface temperature in spring | 1.20 | 0.44 |
| Sea surface temperature in summer | 1.12 | 20.21 |
| Seasonality of precipitation | 1.10 | 11.54 |
| Mean annual temperature | 1.02 | 2.68 |
| Annual precipitation | 0.83 | 1.10 |
| Fall marine primary productivity | 0.41 | 3.35 |
| Sea surface temperature in fall | 0.11 | 4.34 |
| Winter marine primary productivity | 0 | 0 |

**Table 3** Probability of occurrence ranges of the Peruvian pelican (*Pelecanus thagus*) expressed in surface area, and those projected to 2100 under two climate change scenarios.

| Probability of occurrence | Projected surface (km²) | 2.6 rcp scenario | | 8.5 rcp scenario | |
|---|---|---|---|---|---|
| | | km² | Delta (km²) | km² | Delta (km²) |
| 0.089–0.17 | 111,147 | 115,583 | 4,436 | 160,747 | 49,600 |
| 0.17–0.26 | 109,380 | 92,394 | −16,986 | 99,147 | −10,233 |
| 0.26–0.35 | 80,529 | 77,101 | −3,428 | 63,727 | −16,802 |
| 0.35–0.44 | 58,849 | 62,374 | 3,525 | 53,352 | −5,497 |
| 0.44–0.53 | 92,290 | 79,882 | −12,408 | 62,344 | −29,946 |
| 0.53–0.62 | 50,623 | 55,252 | 4,629 | 71,301 | 20,678 |
| 0.62–0.71 | 44,424 | 47,008 | 2,584 | 36,534 | −7,890 |
| 0.71–0.80 | 32,837 | 35,903 | 3,066 | 55,481 | 22,644 |
| 0.80–0.89 | 16,674 | 27,161 | 10,487 | 21,059 | 4,385 |
| **TOTAL** | **596,753** | **592,657** | **−4,096** | **623,692** | **26,939** |

moderate scenario of 2.6 rcp an increase (8.77%) in pelican breeding distribution is predicted (Table 4). Under the severe scenario of 8.5 rcp, a decrease (−19.30%) in pelican breeding distribution is predicted (Table 4). Under the severe scenario, the model predicts a decrease of occurrence probability of nesting sites of the pelican in northern Ecuador and north-central Chile (Fig. 2).

## DISCUSSION

The potential geographic distribution of the pelican currently attains an approximate area of 466,836 km², distributed latitudinally from southern Ecuador (2°13′09″S) to the Taitao Peninsula in southern Chile (46°59′07″S). While, the potential breeding

**Table 4   Probability of occurrence ranges of nesting sites of the Peruvian pelican (*Pelecanus thagus*) expressed in surface area, and those projected to 2100 under two climate change scenarios.**

| Probability of occurrence | Projected surface (km²) | 2.6 rcp scenario | | 8.5 rcp scenario | |
|---|---|---|---|---|---|
| | | km² | Delta (km²) | km² | Delta (km²) |
| 0.1–0.2 | 75,037 | 103,148 | 28,111 | 88,676 | 13,639 |
| 0.2–0.3 | 61,766 | 49,407 | −12,359 | 37,961 | −23,805 |
| 0.3–0.4 | 51,566 | 63,245 | 11,679 | 55,694 | 4,128 |
| 0.4–0.5 | 28,432 | 31,296 | 2,864 | 41,435 | 13,003 |
| 0.5–0.6 | 61,167 | 28,232 | −32,935 | 30,496 | −30,671 |
| 0.6–0.7 | 102,422 | 110,200 | 7,778 | 86,676 | −15,746 |
| 0.7–0.8 | 55,250 | 88,326 | 33,076 | 10,622 | −44,628 |
| **TOTAL** | **435,640** | **473,854** | **38,214** | **351,560** | **−84,080** |

distribution of the pelican currently attains an approximate area of 435,640 km². The mean daytime temperature range and marine primary productivity explain the current potential distribution and breeding of the pelican, which is an endemic species closely associated with the oceanographic barriers of the Humboldt Current Ecosystem (*Jeyasingham et al., 2013*; *Kennedy et al., 2013*). In South America, the Humboldt Current encompasses the greater part of the Pacific coast. Despite the wide latitudinal gradient, the marine–coastal area exhibits a mean daytime temperature range between 4 °C and 8 °C. This is consistent with the highest probability of occurrence of the pelican (https://climatologia.meteochile.gob.cl/application/). In turn, marine productivity is the main predictor of biodiversity and especially of the presence of top predators such as seabirds (*Wakefield, Phillips & Matthiopoulos, 2009*). In the case of the pelican, there is an overlap between areas with high summer marine primary productivity and areas with nesting sites.

Under the future climate change scenarios, the spatial distribution of the pelican is predicted to slightly change. The pelican's breeding distribution might be facilitated by the moderate scenario, increasing near 9%. However, under the severe scenario, the prediction decreased to near −20%. This trend is similar to other studies described for seabirds, whose breeding distribution will be reduced by climate change (*Jenouvrier et al., 2014*; *Krüger et al., 2018*). This increase in habitable area is explained by the climatic conditions in southern Chile, and those climatic conditions will likely be similar to the current conditions of the central coast of Chile (*Falvey & Garreaud, 2009*; *Garreaud, 2011*). Over the last decade, an increase in pelican abundance has been reported along the coast of southern Chile, with observations of large flocks following schools of pelagic fishes in the inner sea (*Imberti, 2005*; *Häussermann, Forsterra & Plotnek, 2012*; *Cursach, Rau & Vilugrón, 2016*; *Cursach et al., 2018*). In this area, there has even been one report of an unsuccessful attempt to nest (*Cursach, Rau & Vilugrón, 2016*). The occurrence of competitive interactions with other seabirds has also been observed with endemic species from Patagonia (*Cursach, Rau & Vilugrón, 2016*). In southern Chile, a group of pelicans was observed displacing nesting pairs of Imperial shag (*Phalacrocorax atriceps*), causing the abandonment of the nest (*Cursach, Rau & Vilugrón, 2016*).

The present study is one of only a few evaluations of the potential effects of climate change on seabirds on the Pacific coast of South America. To evaluate the different scenarios caused by climate change on the spatial distribution of the pelican, we did not include oceanographic variables. This is because the climatic variable "Mean daytime temperature range" was what largely explained the potential spatial distribution and breeding of the pelican. However, further studies are required to assess the effects of climate change on seabird populations, including oceanographic variables. In addition, it is important to recognize that the species spatial distribution models have methodological constraints, including operating based on climatic variables without integrating ecological interactions (*Soberón, Osorio-Olvera & Peterson, 2017*). The co-occurrence of fishing exploitation and El Niño events generates synergistic ecological effects that may push the pelican to critical levels of abundance (*Passuni et al., 2016*; *Barbraud et al., 2018*). In addition, the human disturbances on nesting sites are a key factor in the pelican population dynamics (*Coker, 1919*; *Figueroa & Stucchi, 2012*). Future modeling analyses should include field data about fishing, aquaculture, ENSO events, and human disturbances in nesting sites of the pelican.

In conclusion, the current potential geographic distribution of the pelican is influenced to a large extent by thermal conditions and primary productivity. Under the future climate change scenarios, the spatial distribution of the pelican is predicted to slightly change. The range of breeding distribution of the pelican will be decreased as the main cause of climate change. Under a moderate scenario, we predict that the coasts of southern Chile will constitute an important refuge for the conservation of the pelican. It is necessary that future investigations evaluate in detail the ecological interactions of the pelican and its population increase in southern Chile, considering the different dimensions of the local socio-ecological system.

## ACKNOWLEDGEMENTS

To the University of Los Lagos for financing the costs of this publication. To Dr. Cristián Hernández, Dr. Matthew Lee, Dr. Guillermo Luna-Jorquera, and an anonymous reviewer of PeerJ for their revisions to the manuscript.

### Funding
Jaime Ojeda was supported by CONICYT PIA SUPPORT CCTE AFB170008. The funders had no role in study design, data collection and analysis, decision to publish, or preparation of the manuscript.

### Grant Disclosures
The following grant information was disclosed by the authors:
CONICYT PIA SUPPORT CCTE: AFB170008.

### Competing Interests
Aldo Arriagada is an employee of Litoral Austral Ltda., Chile.
## Author Contributions

- Jaime A. Cursach and Aldo Arriagada conceived and designed the simulations, performed the simulations, analyzed the data, contributed reagents/materials/analysis tools, prepared figures and/or tables, authored or reviewed drafts of the paper, approved the final draft.
- Jaime R. Rau conceived and designed the simulations, analyzed the data, authored or reviewed drafts of the paper, approved the final draft.
- Jaime Ojeda analyzed the data, authored or reviewed drafts of the paper, approved the final draft.
- Gustavo Bizama and Anderson Becerra performed the simulations, contributed reagents/materials/analysis tools.

## Data Availability

Location data of pelican sightings are available as Supplemental Files.

## Supplemental Information

Supplemental information for this article can be found online at http://dx.doi.org/10.7717/peerj.7642#supplemental-information.

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
