# Peer review of "Predicting the potential distribution of the endemic seabird Pelecanus thagus in the Humboldt Current Large Marine Ecosystem under different climate change scenarios"

_PeerJ, doi:10.7717/peerj.7642_

## Round 0.1 · original submission · Major Revisions

Thank you for an interesting paper. It has received two critical and detailed reviews and these should greatly assist with a revision prior to consideration for publication. The reviewers address two major issues that need to be addressed. Reviewer 1 provides keen insight into the seasonality of location data used to describe pelican points for modeling. As these are seasonal, I concur with Reviewer 1 that the work would need to be performed on filtered data, selecting a specific resting or nesting behavior and revising the methods to clarify the selection and choice.

Reviewer 2 noted that despite the stated importance of oceanic variables, they were omitted from the study. This should be corrected and those variables that are available should be used. Please see the comments from Reviewer 2 for a list of reference to consider for data sources.

Overall the manuscript can be very valuable and the reviewers and I agree it is publishable with revision. Please take advantage of these well crafted comments. As the work was performed in MaxEnt, revising the analysis should not be overwhelming and will greatly improve the paper. I look forward to the revision.

·

Basic reporting

No comment

Experimental design

I think that the analysis of the potential distribution in climate change scenarios suffers from a previous analysis about the behaviour and reproductive biology of the pelicans. When you visit the coast of Chile or Peru you can notice that there is a seasonal pattern in the distribution of pelicans, which I think cannot be ignored in this study. I think that this underlying problem leads to the formulation of a hypothesis that is actually applicable to a wide variety of organisms, and the question arises why you are studying the Peruvian pelican? Why do you think this is a good model to investigate the effects of climate change? In addition, the distinctive and new in this study could be that it considers the behaviour and seasonality in the pelican's biology and not only a collection of sightings recording the presence of individuals in sites not categorized by their use. That approach can provide relevant information about future changes in the reproductive distribution of this species, which could be useful for planning the present and future uses of the coastal zones.

Validity of the findings

The results you presented here are representative of the known geographical distribution of the Peruvian pelican. However, for a species like this that during the reproductive season concentrates in some areas along the Peruvian and Chilean coast, these results are confounding. You mentioned that the information comes from sightings, but it is not indicated which of the points included in the analysis corresponding to nesting colonies or resting places. Normally resting places are occupied by pelicans during the non-breeding period. After reproduction, the pelicans disperse along the coast and arrive at places they use to rest, but they also feed on discards in fishing coves. This last behaviour could generate a very important bias in the distribution when considering sighting records that come from fishing terminals. Thus, I believe that it is absolutely necessary to filter the data considering the behaviour of the pelicans and to focus the predictions and the analysis using confirmed information regarding the breeding colonies. This latter approach could reveal more interesting patterns and predictions, given that the reproductive value of the species depends fundamentally on its ability to find suitable sites for nesting near feeding sources.

Additional comments

This is a very interesting study that without a doubt would contribute to improving the knowledge of the ecology of the birds of the Humboldt Current System, especially in the case of endemic species. I am convinced that authors can overcome methodological issues that generate doubts in their results and conclusions. For example, to describe and characterize the oceanographic habitats, you can obtain environmental variables from the Aqua MODIS (Moderate Resolution Imaging Spectroradiometer) satellite. Finally, I want to encourage the author to continue working on this interesting article that will be very useful to better understand the effects of climate change.

Reviewer 2 ·

Basic reporting

The manuscript is well written using clear language throughout. However, more detail should be provided about the data sources for both occurrence and environmental data, and how they entered the species distribution model. Also, Figure 2 is hard to interpret in its current form. My suggestions are as follows

(1) I think a better contextualisation on SDMs for highly mobile pelagic species in the face of both measurement uncertainty and SDM model uncertainty would be good. See e.g. Quillfeld et al. 2017 https://doi.org/10.1111/jav.01238 . In particular it would be helpful to provide more detail on how the occurrence records arose (i.e. at-sea sightings, tracking, etc.)

(2) the occurrence data (1967-2015) and the oceanographic data (2004-2013) cover diufferent time ranges. Please give more detail on how this was dealt with when fitting the model.

(3) In the dioscussion you touch on the topic of seabird-fisheries interactions in the face of ENSO variability. Much work has been done on this topic since the cited works from the 1980's which would enrich the discussion (see e.g. Passuni et al 2016 https://doi.org/10.1890/14-1134.1, Barbraud et al. 2018 https://doi.org/10.1111/ecog.02485).

(4) Figure 2 as presented currently does a poor job of communicating the central message of the manuscript (potential changes in species distribution). I would recommend to redraw it showing relative change to the current distribution in panels B and C rather than absolute occurrence probabilities.

Experimental design

The research question is in principle meaningful, however, I am not convinced by two key modelling decisions:

(1) the authors decided to not including oceanographic variables in the climate forecast scenarios, despite their own results which show the importance of primary production in particular (Table 2). The statement by the authors on lines 124-125 that such projections are not available is plainly wrong. Projections of net primary productivity, sea surface temperature and other oceanographic variables under different IPCC scenarios are available (see e.g. Bopp et al. 2013 doi:10.5194/bg-10-6225-2013), and have been used in other modelling studies projecting distributional change of marine species (e.g. Letessier et al. 2011 https://doi.org/10.1093/plankt/fbr033, Jones & Cheung 2015 doi:10.1093/icesjms/fsu172)

(2) the authors decided model only the impact of mean climate change, despite the fact that the ecosystem at hand is subject to strong shorter-term climate variability as a result of ENSO. While projections of future ENSO effects are arguably less certain than those of the mean climate, I would have liked to see at least a contextualisation of the magnitudes of effects on those two time scales. I.e. how does the predicted mean temperature/primary prodcution/etc for 2100 compare to the extremes of current ENSO cycles.

Validity of the findings

As stated above - while I belive the findings are technically sound given the chosen model and input data, I am sceptical that the model is adequate for its purpose.

Much of this (e.g. the importance of shorter term variability through ENSO, the importance of using oceanographic variables, the interactions between seabirds and fisheries) is touched on in the discussion, and while I don't think it is realistic to model all of these factors, I do feel that some of them should have been included in the actual analysis and not just as discussion points.

---

## Round 0.2 · Minor Revisions

Based on the feedback of a reviewer and the revisions made, I think this paper is nearly ready for publication, please note revisions from Reviewer 2 should be used to make minor revisions and resubmit. This should be easy to work through. Well done.

Reviewer 2 ·

Basic reporting

The basic reporting is appropriate, but the text added in revision requires copy editing, in particular there I find the use of the term "reproductive distribution (lines 28; 35; 69; and several other places) confusing, and wouls suggest replacing it with "breeding distribution" or "breeding season distribution" throughout.

The added sections lines 100-103; 187-190; 228-232; 253-258 require copy editing.

Experimental design

The authors did not fundamentally revise their analysis since the initial submission. As such my assessment of the analysis remains unchanged.

Validity of the findings

The findings are technically sound given the model and input data, but I believe the conclusions that can be drawn from this are limited given the model is a very reduced representation of the study system.

Additional comments

The authors have addressed most of the issues raised about the presentation, but have not revised their analysis approach.

The findings are technically sound given the model and input data, and the write up is adequate (and honest about the model limitations) - so I believe the paper fulfills the PeerJ criteria - but requires some minor copy editing.

---

## Round 0.3 · accepted · Accept

Thank you and congratulations. I appreciate the effort to revise the manuscript and it is ready for publication. Good work.